# Polyethylene Glycol Loxenatide Accelerates Diabetic Wound Healing by Downregulating Systemic Inflammation and Improving Endothelial Progenitor Cell Functions

**DOI:** 10.3390/ijms26052367

**Published:** 2025-03-06

**Authors:** Zerui Ding, Chunru Yang, Xiaojun Zhai, Yuqi Xia, Jieying Liu, Miao Yu

**Affiliations:** 1Key Laboratory of Endocrinology National Health Commission, Department of Endocrinology, Peking Union Medical College Hospital, Chinese Academy of Medical Sciences and Peking Union Medical College, Beijing 100730, China; s2023001082@pumc.edu.cn (Z.D.); yangchunru18@163.com (C.Y.);; 2Center for Biomarker Discovery and Validation, National Infrastructures for Translational Medicine (PUMCH), Institute of Clinical Medicine, Peking Union Medical College Hospital, Chinese Academy of Medical Sciences and Peking Union Medical College, Beijing 100730, China

**Keywords:** diabetes, glucagon-like peptide-1 receptor agonist, endothelial progenitor cell, mitochondria, autophagy

## Abstract

Diabetes wound healing presents several significant challenges, which can complicate recovery and lead to severe consequences. Polyethylene glycol loxenatide (PEG-loxe), a long-acting glucagon-like peptide-1 receptor agonist (GLP-1RA), shows cardiovascular benefits, yet its role in diabetic wound healing remains unclear. Diabetic mice received PEG-loxe (0.03 mg/kg/week, i.p.) for three months. Glucose metabolism was evaluated using the insulin tolerance test (ITT) and oral glucose tolerance test (OGTT). Wound closure rates and angiogenesis-related proteins were analyzed. Serum proteomics was performed using the Olink assay to evaluate systemic inflammation. In vitro, human endothelial progenitor cells (EPCs) were exposed to high glucose and palmitic acid, with or without PEG-loxe treatment. EPC tube formation and migratory capacity were evaluated using the tube formation assay and migration assay, respectively. Levels of nitric oxide (NO) and phosphorylated endothelial nitric oxide synthase (p-eNOS) were quantified. Mitochondrial reactive oxygen species (ROS) production and mitochondrial membrane potential were assessed using MitoSOX and JC-1 staining. Cellular respiratory function was analyzed via the Seahorse XF assay. Autophagy was evaluated by examining the expression of autophagy-related proteins and the colocalization of mitochondria with lysosomes. PEG-loxe improved glucose tolerance, accelerated wound closure, and upregulated the hypoxia-inducible factor-1α/vascular endothelial growth factor/stromal cell-derived factor-1 axis (HIF-1α/VEGF/SDF-1) in diabetic mice. Serum proteomics revealed reduced pro-inflammatory markers and elevated anti-inflammatory IL-5. In vitro, PEG-loxe restored EPC function by enhancing NO production, reducing mitochondrial ROS, improving cellular respiratory function, and restoring autophagic flux. These findings suggest that PEG-loxe offers therapeutic benefits for diabetic wound healing by downregulating systemic inflammation, enhancing angiogenesis, and improving mitochondrial quality control in EPCs, highlighting GLP-1RAs as potential therapies for diabetic vascular complications.

## 1. Introduction

Diabetes is a leading cause of morbidity and mortality worldwide, primarily due to its vascular complications, which place a significant burden on the healthcare system [1]. Many of these complications arise from perturbations of the delicate balance between the vessel growth, maturation, and quiescence [2]. Abnormal wound angiogenesis can result in limb ischemia, hinder the wound-healing process, lead to chronic ulceration, necessitate amputations, and even result in death [3]. To alleviate the burden of diabetes, it is crucial to gain a more comprehensive understanding of diabetic vascular dysfunction and to develop strategies for monitoring and regulating neovascularization.

Among the therapeutic strategies for diabetes, glucagon-like peptide-1 receptor agonists (GLP-1RAs) have emerged as a relatively novel and highly effective class of anti-diabetic medications. These agents have been shown to provide cardiovascular benefits in addition to their glycemic control effects [4]. Recent studies suggest that GLP-1RAs may exert protective effects on vascular function, contributing to improved endothelial health and reduced atherosclerotic risk [5]. The molecular mechanisms underlying these vascular effects appear to involve the modulation of various signaling pathways. GLP-1RAs enhance nitric oxide (NO) bioavailability, leading to vasodilation and improved blood flow [6]. Additionally, GLP-1RAs can regulate both local and systemic metabolism while eliciting anti-inflammatory effects in the cardiovascular system [7]. Polyethylene glycol loxenatide (PEG-loxe) is a novel GLP-1RA with a longer half-life developed by modifying its chemical structure that showed cardiovascular protective effects in diabetes patients [8]. However, the effects and underlying mechanisms of PEG-loxe on diabetic wound healing remain largely unexplored.

Recent studies have suggested that GLP-1RAs are associated with higher circulating levels of endothelial progenitor cells (EPCs) and can improve EPC dysfunction in diabetes by modulating inflammation, reducing oxidative stress, and alleviating mitochondrial dysfunction [9,10,11]. EPCs have garnered attention as both a biomarker of cardiovascular dysfunction in diabetes and a promising mechanism for addressing angiogenic dysfunction, paving the way for strategies aimed at monitoring and regulating neovascularization [12,13]. EPCs are mononuclear cells derived from the bone marrow that can home in on sites of active angiogenesis in animal models of ischemia and contribute to vascular repair [14]. In the event of vascular damage, EPCs are attracted to the ischemic site by cytokine gradients where they promote the angiogenesis process. Two main types of EPCs have been characterized: early EPCs and late EPCs. Early EPCs primarily contribute to tube formation through the secretion of pro-vasculogenic molecules, while late EPCs exhibit a higher capacity for proliferation and play a direct role in the formation of new blood vessels [15,16].

Based on these considerations, we hypothesized that PEG-loxe may enhance wound healing in diabetes by improving overall inflammatory status and exerting a direct effect on EPC functions. We conducted both in vivo and in vitro studies to assess the protective effects of PEG-loxe on diabetic wound healing, its impact on wound angiogenesis, and the role of mitochondria in enhancing EPC function. Our findings aim to provide further insights into the potential therapeutic role of GLP-1RAs in the management of diabetes.

## 2. Results

### 2.1. PEG-Loxe Improved Glucose Metabolism

To evaluate the effect of PEG-loxe on systemic metabolism, body weight data were collected from the initiation of treatment, and ITTs and OGTTs were conducted at 5 and 6 weeks, respectively. Diabetic mice exhibited increased body weight, while PEG-loxe treatment resulted in a slight reduction in weight (Figure 1A). Insulin tolerance tests (ITTs) and oral glucose tolerance tests (OGTTs) revealed impaired glucose clearance in *db*/*db* mice, as evidenced by elevated blood glucose levels at 60 and 120 min post-challenge (*p* < 0.01 vs. *db*/m controls). PEG-loxe treatment markedly improved insulin sensitivity, reducing the area under the curve (AUC) (*p* < 0.001; Figure 1B–E). These results suggest that PEG-loxe effectively improved the glucose metabolism of diabetic mice.

### 2.2. PEG-Loxe Accelerated Diabetic Wound Healing in db/db Mice

To investigate the role of PEG-loxe in diabetic wound healing, a wound model was established by creating dorsal skin wounds in *db*/*db* and control mice. Wound closure rates were measured every other day up to day 10 [17]. Diabetic mice exhibited delayed wound healing compared to the control group, while PEG-loxe treatment notably accelerated wound closure (Figure 1F,G). Hypoxia-inducible factor-1 alpha (HIF-1α), vascular endothelial growth factor (VEGF) and stromal cell-derived factor-1 (SDF-1) are critical mediators of vascular regeneration. The HIF-1α/VEGF/SDF-1 pathway has been shown to facilitate EPC mobilization and accelerate neovascularization [18]. HIF-1α is essential for the expression of SDF-1, which in turn promotes the homing of EPCs to ischemic regions [19]. In diabetic conditions, circulating levels of EPCs and plasma concentrations of SDF-1 are reduced [20]. Animal and in vitro studies have demonstrated that the HIF-1α/VEGF/SDF-1 axis exhibits impaired responsiveness in diabetic contexts, which subsequently hampers neovascularization [18,21]. Consistent with prior studies, the levels of HIF-1α, VEGFA, and SDF-1 in diabetic wound tissues were significantly lower than in controls but were notably upregulated following PEG-loxe treatment (Figure 1H).

### 2.3. PEG-Loxe Regulated Systemic Inflammatory Response During Diabetic Wound Healing

Using Olink proteomics, we analyzed serum protein expression during wound healing. A heatmap of all differentially expressed proteins is presented in Figure 2A. We identified a total of four differentially expressed proteins between the *db*/*db*+PEG-loxe and untreated *db*/*db* groups: Yes1 and Itgb1bp2 were downregulated, while IL-5 and Adam23 were upregulated in the PEG-loxe-treated group (Figure 2B). Notably, glucagon levels were significantly elevated in the *db*/*db* group compared to the control group. This upregulation was attenuated by the administration of PEG-loxe, which is consistent with the drug’s pharmacological mechanism of action in lowering glucagon levels (Figure 2C).

To gain deeper insights into the systematic inflammatory alteration, we compared the expression levels of inflammation-related proteins across the three groups (Figure 2C). The results demonstrated that levels of key inflammatory factors, such as CCL2 (MCP-1), CCL20 (MIP-3α), and ERBB4, were elevated in diabetic mice compared to the normal control group. This is aligned with previous findings in *db*/*db* mice, indicating dysregulation of the inflammatory response [22,23]. Notably, most pro-inflammatory proteins in the *db*/*db* group were upregulated, and were downregulated following PEG-loxe treatment, although some changes were not statistically significant.

These findings suggested that PEG-loxe may accelerate wound healing in diabetic mice, likely by promoting neovascularization and modulating the inflammatory response toward a more regulated state.

### 2.4. PEG-Loxe Protects Endothelial Progenitor Cell Function from Glucotoxicity and Lipotoxicity In Vitro

EPCs play a vital role in neovascularization and wound healing. To evaluate the effect of PEG-loxe on neovascularization, we conducted in vitro experiments to assess EPC function. To simulate the hyperglycemic and lipotoxic environment characteristic of diabetes, EPCs were cultured in high-glucose and high-fatty-acid conditions (HG/FA group). We assessed tube formation and migration using Matrigel and transwell assays, respectively. The results indicated that exposure to high glucose and high fatty acid levels significantly impaired tube formation and diminished migratory abilities compared to untreated control cells (Figure 3A–D). However, when treated with PEG-loxe (HG/FA+PEG-loxe group), both tube formation and migration capacities were partially restored, suggesting that PEG-loxe mitigates the effects of glucotoxicity and lipotoxicity, thereby preserving EPCs’ neovascularization potential.

### 2.5. PEG-Loxe Increased NO Bioavailability in Endothelial Progenitor Cells Under High-Glucose and -Fatty-Acid Conditions

EPC dysfunction in diabetes is associated with reduced nitric oxide (NO) bioavailability, primarily attributed to endothelial nitric oxide synthase (eNOS) uncoupling [24]. Phosphorylation of eNOS at Ser1177 represents an important post-translational modification required for its enzymatic activation. In the present study, EPCs exposed to high-glucose and -fatty-acid (HG/FA) conditions exhibited significantly decreased NO levels in culture supernatants compared to the control group (Figure 3E), a finding consistent with eNOS uncoupling. Notably, treatment with PEG-loxe markedly restored NO production in HG/FA-stimulated EPCs. However, Western blot analysis revealed no statistically significant differences in the phosphorylated eNOS (Ser1177)/total eNOS ratio among experimental groups (Figure 3F,G). These data indicated that PEG-loxe ameliorates impaired NO bioavailability in diabetic EPCs under glucolipotoxic stress, potentially through mechanisms independent of Ser1177 phosphorylation-dependent eNOS activation.

### 2.6. PEG-Loxe Reduces Reactive Oxygen Species and Restores Mitochondrial Membrane Potential in Endothelial Progenitor Cells

The elevated ROS can lead to eNOS uncoupling, promoting a cycle of oxidative stress and endothelial damage [25]. Mitochondria serve as the primary source of cellular ROS, and an increase in ROS within mitochondria can lead to a loss of mitochondrial membrane potential. To explore the role of mitochondrial ROS in EPC dysfunction, we employed MitoSOX staining to assess mitochondrial superoxide production in EPCs. Our results demonstrated that exposure to high glucose and fatty acids (HG/FAs) significantly increased mitochondrial ROS production. In contrast, PEG-loxe treatment significantly reduced HG/FA-induced ROS accumulation (Figure 4A,C). Furthermore, we observed that the mitochondrial membrane potential was significantly diminished under HG/FA conditions, but it was restored upon PEG-loxe treatment, as assessed by JC-1 staining (Figure 4B,D).

### 2.7. PEG-Loxe Restored Mitochondrial Respiratory Function

A Seahorse assay was employed to evaluate whether PEG-loxe could prevent mitochondrial dysfunction in EPCs. The results indicated that mitochondrial respiration rates, including both basal and maximal respiration, were significantly reduced in the HG/FA group. However, treatment with PEG-loxe enhanced the mitochondrial oxygen consumption rate (OCR) in EPCs (Figure 4E–I). These findings, alongside the restoration of mitochondrial membrane potential, suggest that oxidative stress impairs mitochondrial respiratory function. Additionally, PEG-loxe treatment effectively preserved mitochondrial membrane potential and reversed the impairment of mitochondrial respiration.

### 2.8. PEG-Loxe Facilitated Mitochondria Quality Control

Building on the results mentioned above, maintaining mitochondrial homeostasis is essential for the overall health of EPCs. Autophagy is a crucial cellular process that plays a significant role in regulating mitochondrial quality. Impaired mitochondria are tagged and then delivered to lysosomes for degradation, ensuring that the quantity and quality of mitochondria are preserved [26]. To evaluate the induction and progression of autophagy, we analyzed the levels of the proteins LC3 || and p62. In control EPCs, the baseline levels of LC3 || and p62 were low; however, these proteins were significantly increased in response to HG/FA treatment (Figure 5A,C,D), consistent with autophagic flux impairment. In comparison to the HG/FA group, PEG-loxe treatment resulted in a reduction in p62 expression and a tendency toward lower levels of TOM20, a mitochondrial membrane protein (Figure 5B,E). This indicates improved autophagic flux and mitochondrial turnover.

To further investigate the interaction between mitochondria and lysosomes in EPCs, we performed a colocalization assay using MitoTracker and LysoTracker. The yellow dots in the merged images indicate the fusion of red mitochondria with green lysosomes (Figure 5F,G). The results demonstrated impaired fusion of mitochondria and lysosomes under HG/FA conditions. In contrast, treatment with PEG-loxe led to an increase in the number of mitolysosomes, suggesting enhanced mitophagy. Western blot analysis confirmed that, in the HG/FA group, levels of LC3 || and p62 levels did not further accumulate when treated with bafilomycin A1 (Figure 5H), a lysosomal inhibitor. However, PEG-loxe treatment reversed this effect, indicating a restoration of autophagic activity.

Collectively, PEG-loxe treatment partially restored autophagic flux and improved mitochondrial quality control that were impaired by glucolipotoxicity.

## 3. Discussion

Wound healing is an integrative process that requires the coordination of multiple biological events, including the recruitment of immune cells, formation of granulation tissue, vascular growth, and collagen deposition. In diabetes, however, the wound-healing process is often delayed due to various factors, such as inappropriate inflammation response, insufficient pro-angiogenic and vascular maturation factors, and decreased levels of EPCs [3]. The present study demonstrates that PEG-loxe accelerates diabetic wound healing through dual mechanisms: systemic anti-inflammatory regulation and direct improvement of EPC functions.

Inflammation and angiogenesis are closely linked, and the resolution of inflammation promotes wound healing and accelerates tissue repair [27]. Previous research has shown that transient inflammation promotes EPC mobilization, whereas chronic or excessive inflammation reduces circulating EPC numbers and impairs vascular repair [28]. GLP-1RAs are known to suppress circulating pro-inflammatory cytokines in humans, which may contribute to cardiovascular benefits [7,29]. In our study, PEG-loxe administration attenuated pro-inflammatory cytokines (e.g., CCL2, CCL20). CCL2, a potent chemoattractant for monocytes and macrophages, is elevated in the wound area of *db*/*db* mice, prolongs inflammation, and delays healing [22,30]. By suppressing CCL2, PEG-loxe likely mitigates macrophage-driven tissue damage, creating a microenvironment conducive to vascular repair. Similarly, CCL20 has been implicated in keratinocyte dysfunction and delayed re-epithelialization [31]. The downregulation of these cytokines aligns with accelerated wound closure in PEG-loxe-treated mice.

Notably, PEG-loxe significantly elevated serum IL-5, a Th2-derived anti-inflammatory cytokine, further supporting a shift toward anti-inflammatory immune responses. While IL-5 is classically associated with eosinophil activation, studies have proposed that Th2 cytokines promote wound healing by shifting macrophage polarization toward an M2 phenotype, which secretes anti-inflammatory factors and pro-angiogenic factors (e.g., IL-10, VEGF, TGF-β) [32,33]. Interestingly, in patients with asthma, elevated levels of IL-5 and circulating EPCs have been observed [34]. Specifically, EPCs derived from asthma patients demonstrate enhanced proliferative capacity and increased integration into endothelial cell tubes [34], leading to the hypothesis that IL-5 may exert a direct influence on angiogenesis by acting upon EPCs. While conflicting reports exist regarding IL-5’s direct angiogenic effects [35], our data suggest that the increase in IL-5 levels is a consequence and manifestation of a regulated systemic immune response. Such systemic anti-inflammatory effects likely synergize with EPC functional recovery to enhance neovascularization [36].

The migration and function of EPCs are vital for effective diabetic wound repair [37]. Previous studies have underscored the importance of activating HIF-1α and its downstream targets, VEGF and SDF-1, in recruiting EPCs to the wound site and promoting the formation of new blood vessels [18,38,39]. Supporting these findings, our study showed that PEG-loxe administration upregulated the HIF-1α/VEGF/SDF-1 axis in the wound area of mice, thereby accelerating angiogenesis and wound healing. Collectively, these data suggest that PEG-loxe fosters a pro-reparative immune environment, enabling efficient EPC recruitment and neovascularization.

EPC dysfunction in diabetes is closely tied to mitochondrial oxidative stress and defective autophagy. In diabetic conditions, eNOS uncoupling results in diminished NO production and elevated ROS levels [40], contributing to reduced EPC numbers and compromised functionality [17,24]. While moderate levels of endogenous ROS are essential for the activation of autophagy and stress resistance [41,42], excessive ROS production or inadequate scavenging can lead to mitochondrial damage, lipid peroxidation, and ultimately, cell death [43,44]. This disrupts cellular homeostasis, impairs the coupling of eNOS, and promotes oxidative stress and endothelial damage [25,45]. Dysregulation of redox signaling can also impair autophagic processes, creating a feedback loop that exacerbates oxidative stress and further compromises mitochondrial function [46].

Our in vitro experiments demonstrated that high glucose and fatty acid (HG/FA) induced mitochondrial ROS overproduction, depolarized mitochondrial membrane potential, and impaired respiratory capacity—hallmarks of glucolipotoxicity. PEG-loxe treatment not only reduced mitochondrial ROS (Figure 4A,C) but also restored basal and maximal respiration rates (Figure 4E–I), indicating improved electron transport chain efficiency. These effects were paralleled by enhanced NO bioavailability (Figure 3F,G), a critical mediator of EPC survival, migration, angiogenic function, and vascular homeostasis [47]. Mechanistically, the restoration of autophagic flux by PEG-loxe (Figure 5) likely plays a pivotal role in mitochondrial homeostasis. Under HG/FA conditions, autophagic flux was stalled at the lysosomal degradation stage (Figure 5H). This mirrors observations in diabetic kidneys, where lipotoxicity disrupts lysosomal acidification and autophagosome-lysosome fusion [48,49]. PEG-loxe partially reversed this defect, increasing mitolysosome formation (Figure 5F,G) and reducing mitochondrial content (Figure 5E), consistent with restored mitophagy. As damaged mitochondria are a major source of ROS, their selective removal via mitophagy would break the cycle of oxidative stress and eNOS uncoupling, thereby preserving EPC function. These findings align with recent studies showing that GLP-1RAs rescue EPC dysfunction through SIRT3/Foxo3 and mitophagy pathways [9,50], further underscoring mitochondrial quality control as a therapeutic target.

This study has several limitations. First, we did not investigate the EPC number alteration and function improvement in vivo following PEG-loxe treatment, due to the limitations of the relatively small blood and tissue volume in mice. Further clinical studies could validate our findings. Second, the restricted scope of the inflammatory molecule panel employed in this study precluded a comprehensive assessment of the full spectrum of cytokines involved in wound healing. Notably, key mediators such as IL-4, IL-13 [51], IL-33 [52], and IL-20 [53,54], which are known to modulate tissue repair mechanisms, may harbor unexplored pathophysiological significance in this context. These findings emphasize the intricate interplay between immune and metabolic pathways in chronic wound pathogenesis, suggesting a multifaceted regulatory network that warrants further exploration.

Emerging evidence indicates that GLP-1RAs enhance tissue repair across diverse preclinical models, independent of diabetic status [55,56,57]. Expanding their application to these contexts could address unmet clinical needs in bridging metabolic dysregulation and impaired tissue regeneration. Additionally, the mitochondrial protective effect of GLP-1RAs as potential therapeutics for diabetes subtypes characterized by mitochondrial dysfunction [58,59,60]. Furthermore, clinical trials should be designed to assess the long-term safety and efficacy of GLP-1RAs in diverse patient populations, including those with mitochondrial dysfunction, to ensure their broad applicability and therapeutic benefits.

## 4. Materials and Methods

### 4.1. Animals

Male 8-week-old *db*/*db* mice for a model of type 2 diabetes mellitus and age-matched male *db*/*m* mice for the control group were purchased from GemPharmatech Co., Ltd. (Nanjing, China). Mice were housed in a temperature-controlled environment (22–24 °C) with a 12 h light/dark cycle and ad libitum access to standard chow and water. The study was approved by the Laboratory Animal Welfare Ethics Committee of the Peking Union Medical College Hospital (XHDW-2023-145).

### 4.2. Insulin Tolerance Test (ITT) and Oral Glucose Tolerance Test (OGTT)

Insulin tolerance tests (ITTs) and oral glucose tolerance tests (OGTTs) were performed at 5 and 6 weeks after initiating PEG-loxenatide treatment, respectively. For ITTs, mice were fasted for 8 h, and then injected intraperitoneally with 1 IU/kg body weight insulin (i.p.).

Blood glucose levels were measured from the tail tip at 0, 15, 30, 60, and 120 min post-injection using a glucometer. For OGTTs, mice were fasted overnight for 16 h and then orally gavaged with 1 g/kg body weight of glucose dissolved in water. Blood glucose was measured at 0, 15, 60, and 120 min.

### 4.3. Mouse Model of Wound Healing and PEG-Loxenatide Administration

After one week of adaptation, mice were divided into three groups: (1) individuals of the *db*/*m* strain as a control group; (2) untreated *db*/*db* mice; and (3) *db*/*db* mice treated with PEG-loxe (0.03 mg/kg/week, intraperitoneal injection). The drug was administered for 12 weeks before establishing a wound model. On the day of wound creation, mice were anesthetized with 3% isoflurane, and their dorsal fur was shaved. A full-thickness, circular wound with a 6 mm diameter was generated on the dorsum using a sterile punch biopsy. Afterward, the mice were housed individually in separate cages.

### 4.4. Wound-Healing Analysis

Photographs of the wounds were taken every two days starting from the day the wounds were created. The wound area was quantified using ImageJ software (version 1.54g), and the wound closure rate was calculated as follows: wound closure rate (%) = (An − A0)/A0 × 100 (A0 = the wound area on the day of wound creation; An = the remaining wound area on the respective day).

Mice were euthanized on day 10 and the full-thickness wound tissues along with a rim of surrounding unwounded skin margins were excised. Tissue samples were quickly frozen in liquid nitrogen and then transferred to −80 °C for storage.

### 4.5. Serum Olink Analysis

Blood samples were collected via cardiac puncture and centrifuged at 3000× *g* for 15 min to isolate serum. Serum proteins from control, diabetic, and PEG-loxe-treated diabetic groups were analyzed using the Olink Mouse Exploratory Panel (Biotree, Shanghai, China). Normalized protein expression (NPX) values were log2-transformed, and differentially expressed proteins were identified using the DEGseq package (version 1.61.0) with a significance threshold of *p* < 0.05.

### 4.6. Cell Culture and Treatment

Human peripheral blood-derived endothelial progenitor cells (EPCs) were purchased from FuHeng Biology Co., Ltd. (Shanghai, China). EPCs were cultured in EGM-2 BulletKit medium (Lonza, Basel, Switzerland) supplemented with recombinant human rhEGF, rhFGF-B, rhVEGF, rhIGF-1, ascorbic acid, heparin, 5% fetal bovine serum, and 1% penicillin/streptomycin. The cells were maintained at 37 °C in a humidified incubator with 5% CO_4_. To mimic diabetic conditions, EPCs were exposed to high glucose (30 mM) and palmitic acid (100 μM) (HG/FA group) for 24 h. For intervention, cells were co-treated with 250 nM PEG-loxe (HG/FA+PEG-loxe group).

### 4.7. NO Detection

The concentration of nitrite, an indicator of nitric oxide (NO) production, was determined using the Griess reaction. The nitrite detection kit (Beyotime, Shanghai, China) was used according to the manufacturer’s instructions. Briefly, 50 µL of supernatant or sodium nitrite (NaNO_2_) standard was mixed with 50 µL of Griess Reagent I and 50 µL of Griess Reagent II in a 96-well plate. After a 10 min incubation in the dark, the NO concentration was quantified by measuring the absorbance at 540 nm using a microplate reader. The NO content was calculated based on the optical density (OD) values, with each group normalized to the control group.

### 4.8. Transwell Migration Assay

EPCs were pretreated for 24 h before the migration assay. Cells were then collected and resuspended in serum-free EBM-2 medium. A total of 2 × 10^4^ EPCs were added to the upper chamber of a 24-well transwell plate (8.0 µm pore size; Corning, New York, NY, USA), and 500 μL of EGM-2 medium containing 5% FBS was added to the lower chamber. Cells were incubated at 37 °C for 8 h. After fixation with paraformaldehyde, the cells were stained with crystal violet solution (Beyotime, Shanghai, China). Images of cells that migrated to the lower chamber were captured using an inverted light microscope (Nikon, Tokyo, Japan), and the number of migrating cells was counted using ImageJ.

### 4.9. Tube Formation Assay

For the tube formation assay, 50 μL Matrigel (Corning, New York, NY, USA) was added to the wells of a 96-well plate and incubated at 37 °C for 30 min for polymerization. EPCs (1.5 × 10^4^ cells per well) were then seeded onto the Matrigel. After 6 h of incubation, images of tube-like structures were captured, and the length of the capillary-like structures was measured using ImageJ software.

### 4.10. Detection of Reactive Oxygen Species (ROS)

Mitochondrial ROS levels were quantified using MitoSOX™ Red (Thermo Fisher Scientific, Inc., Waltham, MA, USA), a fluorescent dye specific for mitochondrial superoxide. EPCs were stained according to the manufacturer’s instructions, and fluorescence images were acquired using a confocal laser scanning microscope (Nikon, Tokyo, Japan).

### 4.11. Mitochondrial Membrane Potential Assay

The mitochondrial membrane potential was examined using JC-1 dye (Beyotime, C2003S), which is sensitive to changes in mitochondrial membrane potential. EPCs were incubated with the JC-1 working solution at 37 °C for 20 min. After treatment, cells were examined by fluorescence microscopy (Nikon AXR inverted confocal microscope). For quantitative analysis, three random fields (1000× magnification) were selected for each group, and the ratio of red to green fluorescence (indicative of mitochondrial membrane potential) was measured and analyzed using ImageJ software.

### 4.12. Colocalization of Mitochondria and Lysosomes

The colocalization of mitochondria and lysosome assay was performed as previously described [61]. In short, EPCs were stained with MitoTracker Deep Red (Invitrogen, Carlsbad, CA, USA, M22426) at a final concentration of 250 nM and LysoTracker Green (Beyotime, Shanghai, China, C1047S) at a final concentration of 50 nM according to the manufacturer’s instructions, and then photographed using a Nikon AXR confocal laser scanning microscope to assess the colocalization of mitochondria and lysosomes.

### 4.13. Seahorse XF Analysis

The oxygen consumption rate (OCR) and extracellular acidification rate (ECAR) were measured using an Seahorse XF96 Analyzer (Agilent, Santa Clara, CA, USA). The experiments were conducted at 37 °C and pH 7.4, using Seahorse XF DMEM medium supplemented with glucose, pyruvate, and glutamine as the detection solution, following the manufacturer’s instructions. All respiratory parameters were corrected for non-mitochondrial respiration and background signals by adding 1.5 µM oligomycin, 2 µM FCCP, and 0.5 µM rotenone/antimycin A (Rot/AA).

### 4.14. Western Blotting

Total protein was extracted from skin tissue or EPCs using RIPA buffer containing protease inhibitors, PMSF, and phosphatase inhibitors. Equal amounts of protein (15–20 µg) were loaded onto a 4–20% SDS-PAGE gel (Boyi, Changzhou, China) and then transferred to PVDF membranes. After blocking with protein-free rapid blocking solution (Servicebio, Beijing, China) for 15 min, the membranes were incubated overnight at 4 °C with primary antibodies against the target proteins HIF-1α (CST, Beverly, MA, USA, 1:1000 dilution), VEGFA (Abcam, Cambridge, UK, 1:1000 dilution), SDF1/CXCL12 (CST, 1:1000 dilution), eNOS (CST, 1:1000 dilution), phosphor-eNOS (CST, 1:1000 dilution), p62 (CST, 1:1000 dilution), LC3A/B (CST, 1:1000 dilution), and β-actin (CST, 1:1000 dilution). The membranes were then incubated for 1 h at room temperature with HRP-conjugated secondary antibodies (Easybio, Zhenjiang, China). Protein bands were detected using a chemiluminescence (ECL) method and band intensities were quantified using ImageJ.

### 4.15. Statistical Analysis

Data were displayed as mean ± SEM. Comparisons between the groups were analyzed by unpaired Student’s *t* tests. Comparisons among multiple groups were analyzed by one-way analysis of variance (ANOVA) followed by Tukey’s post hoc test. The two-way ANOVA was used for analyzing wound closure followed by Dunnett’s post hoc test. A *p*-value < 0.05 was considered statistically significant.

## 5. Conclusions

In conclusion, PEG-loxe accelerates diabetic wound healing through a dual strategy: resolving systemic inflammation to create a pro-angiogenic environment and rectifying mitochondrial dysfunction in EPCs via enhanced autophagic flux, offering new insights into diabetic wound management. Future studies should prioritize clinical translation, focusing on validating EPC-mediated repair in human diabetic wounds.

## Figures and Tables

**Figure 1 ijms-26-02367-f001:**
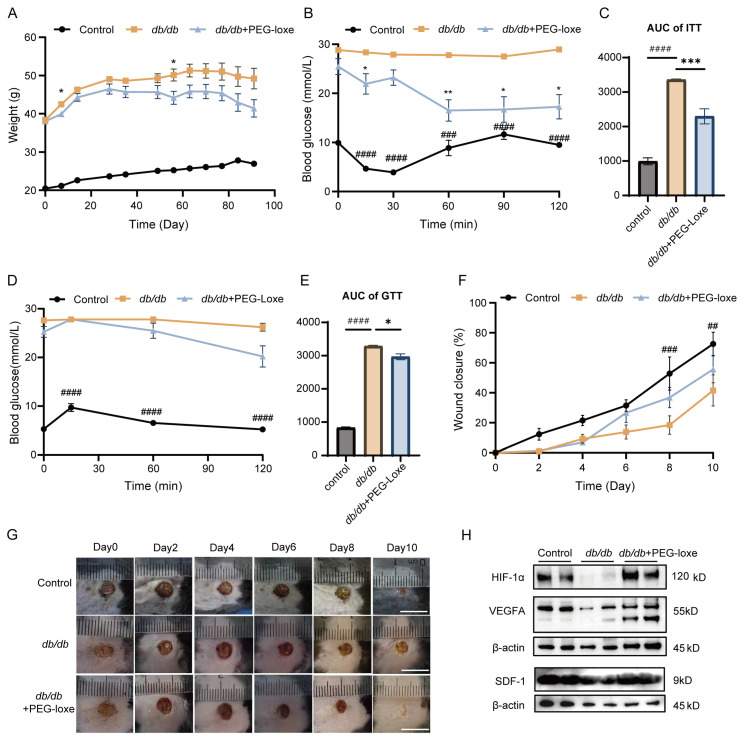
PEG-loxe improved glucose metabolism and accelerated wound healing in diabetic mice. (**A**) Changes in weight of mice over time among each group. (**B**) Insulin tolerance test (ITT) among each group. (**C**) Area under curve (AUC) of ITT. (**D**) Glucose tolerance test (GTT) among groups. (**E**) Area under curve (AUC) of GTT. (**F**) Wound closure rate over time. (**G**) Representative images of wound sites at different time points; scale bar = 1 cm. (**H**) Western blot analysis showing protein expression levels of HIF-1α, VEGFA, and SDF-1 in wound tissues from each group. Control vs. *db/db*: ## *p* < 0.01, ### *p* < 0.001, #### *p* < 0.0001; *db/db*+PEG-loxe vs. *db/db*: * *p* < 0.05, ** *p* < 0.01, *** *p* < 0.001. *n* = 3–6 for each group.

**Figure 2 ijms-26-02367-f002:**
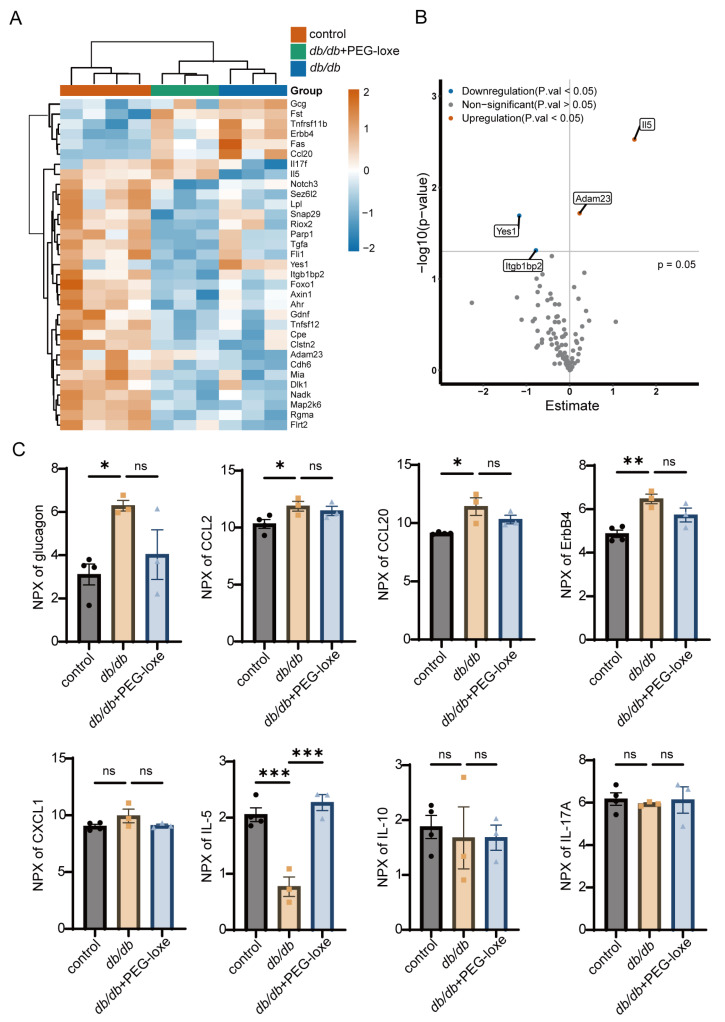
PEG-loxe regulated the systemic inflammatory response during diabetic wound healing. (**A**) Heatmap illustrating the expression patterns of all differentially expressed proteins across groups. (**B**) Volcano plot displaying differentially expressed proteins with *p*-value < 0.05 in the *db*/*db*+PEG-loxe group compared with the *db*/*db* group. (**C**) Comparative analysis of representative inflammation-related factors among groups. NPX, log2-transformed normalized protein expression values. * *p* < 0.05, ** *p* < 0.01, *** *p* < 0.001, ns: not significant; *n* = 3–4 for each group.

**Figure 3 ijms-26-02367-f003:**
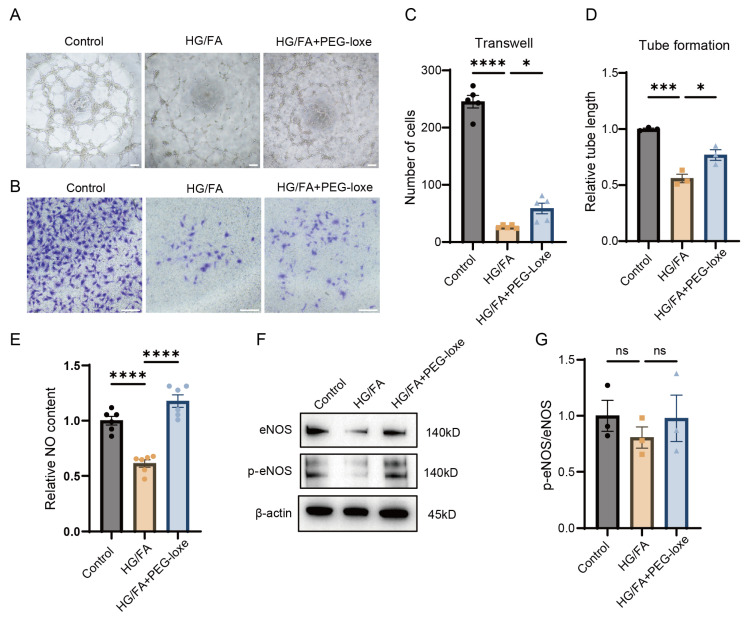
PEG-loxe alleviates EPC dysfunctions induced by high glucose and fatty acids. (**A**) Tube formation assay of EPCs; magnification: ×40; scale bar = 200 μM. EPCs were treated with 30 mM glucose and 100 μM palmitic acid, with (HG/FA+PEG-loxe) or without (HG/FA) 250 nM PEG-loxe for 24 h. (**B**) Transwell migration assay evaluating EPC motility; magnification: ×100; scale bar = 200 μM. (**C**) Quantitative analysis of the tube formation assay. (**D**) Quantitative analysis of the Transwell migration assay. (**E**) Relative nitric oxide (NO) content in the culture medium supernatant of EPCs. (**F**) Western blot analysis of eNOS and phosphorylated-eNOS-Ser-1177 (p-eNOS) expression in EPCs. (**G**) Quantification of p-eNOS/eNOS ratio. * *p* < 0.05, *** *p* < 0.001, **** *p* < 0.0001, ns: not significant; *n* = 3–6 for each group.

**Figure 4 ijms-26-02367-f004:**
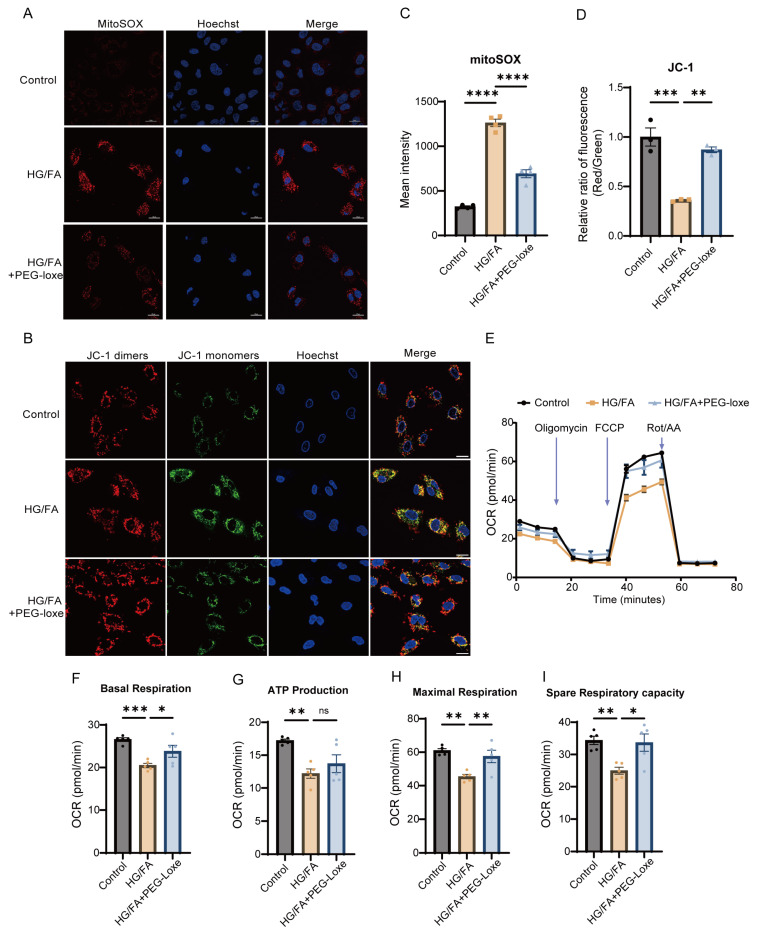
PEG-loxe ameliorates mitochondrial injury and impaired respiration induced by high glucose and fatty acids in EPCs. EPCs were treated with 30 mM glucose and 100 μM palmitic acid, with (HG/FA+PEG-loxe) or without (HG/FA) 250 nM PEG-loxe for 24 h. (**A**) Representative MitoSOX fluorescence images of EPCs showing mitochondrial ROS production upon different treatments; magnification: ×1000; scale bar = 20 μM. (**B**) Mitochondrial membrane potential visualized using JC-1 dye; magnification: ×1000; scale bar = 20 μM. (**C**) Quantitative analysis of red fluorescence intensity from the MitoSOX assay. (**D**) Relative red/green fluorescence ratio from JC-1 staining, indicating changes in mitochondrial membrane potential. (**E**) Mitochondrial oxygen consumption rate (OCR) of EPCs assessed using the Seahorse Mito Stress Test. (**F**) Basal respiration. (**G**) ATP production. (**H**) Maximal respiration. (**I**) Spare respiratory capacity. FCCP, carbonyl cyanide 4-(trifluoromethoxy) phenylhydrazone; Rot, rotenone; AA, antimycin A. * *p* < 0.05, ** *p* < 0.01, *** *p* < 0.001, **** *p* < 0.0001, ns: not significant; *n* = 5 for each group.

**Figure 5 ijms-26-02367-f005:**
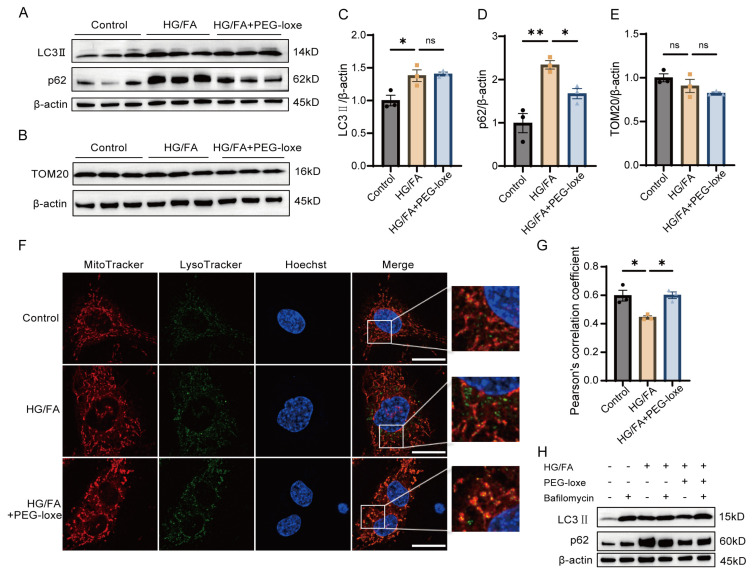
PEG-loxe improved autophagic flux and reduced mitochondrial accumulation in EPCs under high-glucose and -fatty-acid conditions. EPCs were exposed to 30 mM glucose and 100 μM palmitic acid, with (HG/FA+PEG-loxe) or without (HG/FA) 250 nM PEG-loxe for 24 h. (**A**) Western blot analysis of the expression levels of autophagic markers LC3 || and p62, as well as mitochondrial marker TOM20. (**B**) Representative blot images for TOM20 expression. (**C**) Quantification of LC3 || expression. (**D**) Quantification of p62 levels. (**E**) Quantification of TOM20 levels, reflecting mitochondrial content. (**F**) Mitochondria–lysosome colocalization assessed through fluorescence imaging, demonstrating PEG-loxe’s effect on mitophagy; magnification: ×1000; scale bar = 20 μM. (**G**) Quantitative analysis of mitochondria–lysosome colocalization, supporting enhanced mitophagic activity in the PEG-loxe group. (**H**) Western blot analysis of LC3 || and p62 levels treated with or without bafilomycin A1. * *p* < 0.05, ** *p* < 0.01, ns: not significant; *n* = 3 for each group.

## Data Availability

The data used and analyzed during the current study are available from the corresponding author on reasonable request.

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
