# Peer review of "Polyethylene Glycol Loxenatide Accelerates Diabetic Wound Healing by Downregulating Systemic Inflammation and Improving Endothelial Progenitor Cell Functions"

_ijms, 2025, doi:10.3390/ijms26052367_

Round 1
Reviewer 1 Report
Comments and Suggestions for Authors
1. This is a wonderful experimental work with substantial original data. Is PEG-loxe only for the diabetic wound healing, or can the authors extrapolate these results to any wound healing?
2. About the diabetes, especially type 2 diabetes, WFS1 gene is one of the main risk factors in humans. This gene is related to the regulation of growth hormone and to the mitochondrial functions and metabolism (PMID: 19293327, 37759745).
3. What about innate immunity signals in the skin, like GPR15 (PMID: 26348578). GPR15 is upregulated in psoriasis and leads to the wound healing problems in psoriasis, hyperkeratosis and hyperactive skin. The authors mention IL10 and some other cytokines, but GPR15 should also be discussed. And some more psoriasis-related studies like a large transcriptional profiling study (PMID: 25897967) and IL-20 related connections (PMID: 14712309).
Reviewer 2 Report
Comments and Suggestions for Authors
The paper is very clear and consistent. In abstracts, the authors have used abreviations for everything, except for the mediators: HIF-1α, VEGFα and SDF-1. Also, the abreviation used for high glucose and palmitic acid condition (HP) is not didatical.
It is not didatical, to use 8 subchapters (2.1 to 2.8) for results, whereas for discussion, it was used only 2 (3.1 and 3.2). In same way, the results chapters have used a different construction, than the one used for discussions. For instance, for results "2.1. PEG-loxe improved glucose metabolism"active noun + paste tense verb + passive subject". Already for discussion, it were used only nouns, i.e. "Systemic anti-inflammatory regulation and vascular repair"
The conclusion section was missed.
In line 164, I would ask if the notable effect of PEG-loxe on NO levels was statistically significant?
The figures must have the resolution improved
Some few typing mistakes, i.e glued words with [ref], lines 45, 53 and 68.
Round 2
Reviewer 1 Report
Comments and Suggestions for Authors
The authors have addressed all my comments.